# Mortality Risk Prediction in Abdominal Septic Shock Treated with Polymyxin-B Hemoperfusion: A Retrospective Cohort Study

**DOI:** 10.3390/jpm13071023

**Published:** 2023-06-21

**Authors:** Sergio Garcia-Ramos, Estrela Caamaño, Patrocinio Rodríguez Benítez, Pilar Benito, Alberto Calvo, Silvia Ramos, Mercedes Power, Ignacio Garutti, Patricia Piñeiro

**Affiliations:** 1Department of Anaesthesiology and Critical Care Medicine, Gregorio Marañon National Hospital, 28007 Madrid, Spain; sergius_5@hotmail.com (S.G.-R.);; 2Department of Nephrology, Gregorio Marañon National Hospital, 28007 Madrid, Spain

**Keywords:** septic shock, polymyxin, hemoperfusion, endotoxin

## Abstract

Endotoxin, a component of the cell membrane of gram-negative bacteria, is a trigger for dysregulated inflammatory response in sepsis. Extracorporeal purification of endotoxin, through adsorption with polymyxin B, has been studied as a therapeutic option for sepsis. Previous studies suggest that it could be effective in patients with high endotoxin levels or patients with septic shock of moderate severity. Here, we perform a retrospective, single-centre cohort study of 93 patients suffering from abdominal septic shock treated with polymyxin-B hemoperfusion (PMX-HP) between 2015 and 2020. We compared deceased and surviving patients one month after the intervention using X^2^ and Mann-Whitney U tests. We assessed the data before and after PMX-HP with a Wilcoxon single-rank test and a multivariate logistic regression analysis. There was a significant reduction of SOFA score in the survivors. The expected mortality using APACHE-II was 59.62%, whereas in our sample, the rate was 40.9%. We found significant differences between expected mortality and real mortality only for the group of patients with an SOFA score between 8 and 13. In conclusion, in patients with abdominal septic shock, the addition of PMX-HP to the standard therapy resulted in lower mortality than expected in the subgroup of patients with intermediate severity of illness.

## 1. Introduction

Sepsis and septic shock are associated with a high mortality rate. More than one out of four patients die despite appropriate management, including early identification and treatment with broad-spectrum antimicrobials, source control, and goal-directed resuscitation therapies for hemodynamic, respiratory, and metabolic management [1].

The increasing incidence of sepsis and its high economic burden urge us to find new treatment strategies, as the key concepts for sepsis management have remained unchanged over the past 20 years.

Endotoxin, a component of the outer cell membrane of gram-negative bacteria, trigger a dysregulated inflammatory response in sepsis. In patients with abdominal septic shock, high endotoxin activity is associated with poor outcomes [2]. Extracorporeal purification of endotoxin through adsorption has been studied as a potential therapeutic option.

Polymyxin B (PMX) heavily binds to circulating endotoxin [3], rendering PMX hemoperfusion a useful therapy for decreasing circulating endotoxin levels in septic shock.

The evidence for the use of PMX in sepsis remains unclear. The most important randomized controlled trial (RCT) performed to date was unable to find differences in mortality between patients treated with PMX-HP and those treated with standard care [4]. Other cohort studies disagree on the same issue [5]. A recent meta-analysis reported improvements in different outcomes, such as in-hospital stay or ventilator-free days, when using PMX-HP [5].

Considering the conflicting evidence, it is crucial to identify specific patients with sepsis in whom PMX-HP can be beneficial. Previous studies have suggested that it could be effective in patients with higher endotoxin levels [2] or in patients with septic shock and moderate severity [6].

Therefore, this study aimed to analyze the mortality of patients with abdominal septic shock treated with PMX-HP in a postoperative intensive care unit. Furthermore, we aimed to compare the clinical characteristics of deceased patients with those of the survivors.

## 2. Materials and Methods

### 2.1. Design

This retrospective, observational, cohort study was performed at Gregorio Maranon Hospital (Madrid, Spain). The data included patients who were treated with PMX-HP between 2015 and 2020.

The study was approved by the Hospital’s Ethical Committee. Inclusion criteria were:Patients admitted with abdominal septic shock diagnosis, treated with PMX-HP.The intervention included two hemoperfusion treatments of 4 h, within 24 h.Age 18 to 90 years.Available medical records containing all the necessary clinical data.

Exclusion criteria were defined as:Patients with confusing or unavailable medical records.Patients who deceased after a single treatment with polymyxin.Non-abdominal septic shock or multiple septic sources.

Septic shock was defined following the 2016 Third International Consensus (Sepsis-3), specified as a subset of sepsis with a vasopressor requirement to maintain a mean arterial pressure of ≥65 mmHg and serum lactate level greater than 2 mmol/L [7].

The severity of illness was assessed using the adjusted APACHE II [8] and Sequential Organ Failure Assessment (SOFA) [9] scales, both before and after PMX-HP. The expected mortality was calculated using the prognostic value of the APACHE II scale.

The source of infection was diagnosed as the abdomen by physical examination and CT-scan. The term upper abdominal sepsis was used for source infections proximal to the Ligament of Treitz (esophagus, stomach, and duodenum) and referred to lower abdominal sepsis when involving the rest of the small intestine and colon. The third division included the hepatobiliary and pancreatic systems (gallbladder, pancreas, cystic, and common bile duct).

Moreover, we gathered common causes for intestinal pathology as spontaneous intestinal perforation, wound dehiscence, intestinal ischemia, bowel obstruction, inflammation (involving cholecystitis, pancreatitis, and other inflammatory intestinal pathologies), and other exceptions not previously included.

Indications for haemoperfusion were established according to standard clinical practice. The current criteria include: (1) abdominal septic shock, (2) moderate to high vasopressor requirement besides fluid infusion during the first 4 h of treatment, and (3) absence of absolute contraindication for PMX-HP as in the safety data sheet.

Where possible, all available microbiological data were recovered. For the diagnosis, we used intra-operative intra-peritoneal swabs and blood cultures (assuming secondary bacteremia).

All patients received antimicrobials according to the standard ICU policy based on international guidelines [10,11]. Monotherapy and combination antibiotic therapy were used, depending on the risk factors for infection with multidrug-resistant bacteria. The standard treatment always contained beta-lactam antibiotics, except for allergic individuals. We also recovered from the use of antifungal agents.

The surgical team established the need for surgery to control the source of sepsis according to standard clinical practice. The criteria for surgical intervention were evidence of perforation or wound dehiscence, evidence of peritonitis, clinical deterioration with no response to conservative treatment, and the need for exploratory surgery.

The need for renal replacement therapy was established following the usual indications: refractory hyperkalemia, refractory metabolic acidosis, uremic encephalopathy or urea levels > 250 mg/dL, and fluid overload despite diuretics.

### 2.2. Statistical Analysis

Categorical variables are described as frequencies and percentages. Continuous variables are reported as median (interquartile range [IQR]). SPSS V.27 (IBM) was used for statistical analysis. We performed X^2^ test to compare the frequency of categorical variables in the two mortality groups, and the Mann-Whitney U test was used for continuous variables.

To assess the data before and after PMX-HP intervention, we performed a Wilcoxon signed-rank test.

In the multivariate logistic regression analysis, we used clinically relevant variables with a *p* < 0.1 value.

## 3. Results

### 3.1. Patient Selection

A total of 105 patients were registered during the study period. As shown in the flowchart, only 93 patients met the study selection criteria (see Figure 1).

### 3.2. Patient Characteristics

Patient demographics stratified by mortality in the first month are shown in Table 1. The median age was significantly higher in the deceased patients. There were no sex-related differences in the mortality rate.

Among the deceased patients, we found significant differences between patients with renal insufficiency and those on chronic dialysis. There were no differences in other comorbidities, such as arterial hypertension, ischemic heart disease, or peripheral vascular disease.

Regarding the severity of illness, the median APACHE II and SOFA scores were 28 and 13, respectively (Table 1).

### 3.3. Sepsis Source

Most patients presented with a lower gastrointestinal tract (61%), with wound dehiscence being the most common cause of sepsis (33%).

There were no significant differences in death at one month among locations (upper abdominal, lower abdominal, or biliopancreatic), although there was a trend towards higher mortality in cases of intestinal ischemia cases (61.1% mortality rate in the subgroup).

### 3.4. Microbiological Results

Microbiological species were isolated (either by intra-operative fluid culture or blood culture) from 54 patients. Most were Gram-negative bacilli, either alone or in combination with other agents. Mortality was independent of the Bacillus species involved.

### 3.5. Antimicrobials

In general, beta-lactam antibiotics were the preferred choice, especially carbapenems (88.2%). Monotherapy was selected for 44.1% of patients, and two antibiotics were chosen for 43% of patients. The number of antibiotics had no effect on the mortality rate.

### 3.6. Complications Associated with the Use of PMX-HP

In this sample of patients, no evidence of serious complications related to PMX-HP treatment was found.

### 3.7. Treatment Evaluation

To evaluate the results of PMX-HP, we assessed the laboratory tests before and after therapy, the SOFA score, and other clinical variables comparing the two groups of patients, deceased and survivors, after one month. The data are presented in Table 2.

The table displays the relevant pre-filter and post-filter analytical results and their percentage variation for surviving and deceased groups. The p corresponds to the statistical significance of the difference between pre- and post-filter values.

The asterisk (*) symbolizes those analytical values in which there were significant differences when comparing survivors with the deceased.

As shown in the table, there was a significant reduction in the SOFA score in the survivors in contrast to the absence of reduction in the deceased.

Patients who underwent two sessions of PMX-HP no longer needed vasopressors to maintain MAP > 65 mmHg and presented a significantly reduced mortality.

Regarding other variables, the group of survivors showed a clear improvement in laboratory tests after PMX-HP, especially pH, lactate, and INR. There was also a reduction in acute-phase reactants, with a decrease in C-reactive protein level (21.35 mg/dL to 10.85) and procalcitonin (6.75/dL to 2.57 pg/dL).

There was no significant improvement in these variables after PMX-HP in the group of patients who died within the first month.

### 3.8. Mortality Comparison

The expected mortality using the APACHE-II was 59.62%, whereas in our sample, the rate was 40.9%. There was a significant difference with *p* < 0.001.

When analyzing the groups according to SOFA scores, we found significant differences between expected and real mortality only for the group of patients with SOFA scores between 8 and 13. Excluding this group, there were no other differences (Figure 2).

### 3.9. Multivariant Analysis

To carry out the analysis, we included three variables considered the most significant 30-day mortality predictors. As shown in Table 3, the variables associated with a higher mortality rate were chronic kidney disease and increased lactate levels after hemoperfusion. Withdrawal of vasopressors after one session of PMX-HP was associated with a decreased mortality rate.

## 4. Discussion

Our study analyzed the factors affecting mortality in patients treated with PMX-HP. We performed the analysis in a homogeneous group of patients, specifically in those with abdominal septic shock. Despite the existence of some studies focused on abdominal disease, with contradictory results [12,13,14], most of the studies included sepsis with different etiologies.

### 4.1. Severity and Mortality

The sample included patients with severe to very severe abdominal septic shock, with median scores of 28 and 13 on the APACHE II and SOFA severity scales, respectively. The expected mortality rate using the APACHE II score should have been 59.62%, while the one we obtained was 40.9%. Previous studies focused on mortality due to sepsis with ICU admission (without differences according to severity or presence of shock) have shown a global mortality rate around 41.9% [1] Nevertheless, they included a heterogeneous sample of patients with a variety of sepsis sources.

The use of PMX-HP, besides a solid pathophysiology, has yielded contradictory evidence. This fact, along with its high cost, has forced the SSC 2021 guidelines to make recommendations against its use [15]. Alternatively, some meta-analyses have suggested that PMX-HP could be a useful treatment for patients with sepsis [5]. The most important trial to date, EUPHRATES, showed no reduction in mortality in patients treated with PMX-HP [4]. However, a post hoc analysis on the same patients displayed a clinical benefit, with a reduction on the need of vasopressors, ventilator-free days, and mortality in the patients with endotoxin activity measured between 0.6 and 0.89 (intermediate level). Currently, a new clinical trial is being conducted to evaluate the impact of PMX-HP in this subset of patients [16].

Fujimori et al. [17] analyzed PMX-HP in patients with sepsis under real conditions in a nationwide study in Japan. When stratifying according to SOFA score, they found that patients with a score between 7 and 12 benefited from hemoperfusion.

This is especially interesting, since both studies concluded that HP-PMX may not be suitable for all types of patients, but it could be useful in a specific subset of patients. In our study, when we compared the expected mortality according to APACHE II and real mortality, in each severity group according to SOFA, we found statistically significant differences in the patients with intermediate severity (SOFA 8-13), as shown in Figure 2. This finding is consistent with the results of the aforementioned studies. Hence, we can conclude that the filter may be effective in patients with intermediate severity, and not in those who either do not present a very severe condition or whose condition is so severe that PMX-HP does not show beneficial effects.

### 4.2. Renal Failure and Mortality

Regarding the patients’ medical history and mortality, the association between chronic kidney disease and mortality (16% vs. 36.8%) was particularly striking (Table 3), being an independent mortality factor in the multivariate analysis with an OR of 3.6 (Table 3).

In contrast, patients who presented with oliguria also had a higher risk of death (61.8% vs. 86.8%). These data are consistent with those reported in previous studies. For example, Bou Chebl et al. [18] described CKD as the most determining prognostic factor, increasing the mortality rate by up to 50%. Van Herreweghe et al. attempted to determine the predictors of mortality in septic patients with renal failure who required renal replacement therapy (RRT). They only identified the initial SOFA score as an independent predictor of mortality, but found that when sepsis was associated with renal failure requiring RRT, mortality was up to 50.5% [19]. Both our data and theirs suggest that patients with chronic renal disease or those requiring RRT are at special risk; therefore, any patient with suspected sepsis and associated factors should be treated early and aggressively, since these are the patients with the highest mortality.

### 4.3. Type of Pathogen and Mortality

Although PMX-HP has a clear therapeutic target, endotoxin or lipopolysaccharide (LPS), which is mainly found in the walls of Gram-negative bacteria [20,21], there were no differences in mortality according to the type of microorganism involved, whether Gram-negative, Gram-positive, or anaerobic.

These data may be due to the polymicrobial nature of sepsis of abdominal origin [10], when, despite isolating only one type of bacteria, others are actually involved. Another factor that could explain this is bacterial translocation. Numerous studies have shown high levels of LPS, regardless of the type of microorganism, even in Gram-positive sepsis. This is possibly due to intestinal bacterial translocation in patients with shock, thus contributing to increased endotoxin levels and generating a greater inflammatory response [22,23].

### 4.4. Combination Antibiotic Therapy Versus Monotherapy

We found no significant differences in mortality based on the use of monotherapy antibiotics (mostly carbapenems) versus a combination of one or two drugs. In most cases, the second antibiotic used was a glycopeptide (vancomycin) or oxazolidinone (linezolid) as empirical coverage against E. faecium. Although the evaluation of antibiotic use is not the subject of this study, the data are congruent with other articles in which it is preferred to avoid antibiotic combinations, except in specific situations [24,25].

#### Treatment Outcome and Mortality

Another major finding of this study was the relationship between the early response to PMX-HP treatment and mortality in the first 30 days. Patients who showed significant improvement after two doses of PMX-HP were more likely to survive than those who did not. Thus, improvements in pH, HCO3-, and lactate levels were associated with a good prognosis (Table 2). The relationship between procalcitonin reduction and mortality was particularly noticeable. Thus, there was a significantly greater reduction in patients who survived than in those who died. This is congruent with other studies in septic patients in which procalcitonin was associated with a good response to treatment [26].

Another indicator of responsiveness to PMX-HP is the requirement for vasoactive drugs. Patients who underwent two sessions of PMX-HP improved hemodynamically until vasopressor suspension had a longer survival time.

Regarding SOFA reduction, it is shown that those patients with significant SOFA reduction after PMX-HP had a higher chance of survival. In our sample, patients who survived in the first month achieved a significant reduction in SOFA score (from 13 to 11), while those without SOFA reduction died. This parameter, together with the other analytical parameters mentioned above, could be used to assess the utility of early treatment, avoiding overtreatment in patients who may eventually die.

### 4.5. Study limitations

The greatest strength of this study is that it is a homogeneous database in a very specific context, PMX-HP, as a treatment for abdominal septic shock.

The main weakness is the absence of a control group as a result of the use of PMX-HP as standard care in this context since 2015. Furthermore, retrospective studies can be intrinsically biased owing to the challenges of information gathering.

## 5. Conclusions

In patients with abdominal septic shock, the addition of PMX-HP to the standard therapy resulted in lower mortality than expected in the subgroup of patients with intermediate severity of illness in the postoperative Intensive Care Unit. Given the observational nature of this research, it is necessary to conduct randomized controlled trials to determine the role of PMX-HP in abdominal septic shock in surgical patients.

## Figures and Tables

**Figure 1 jpm-13-01023-f001:**
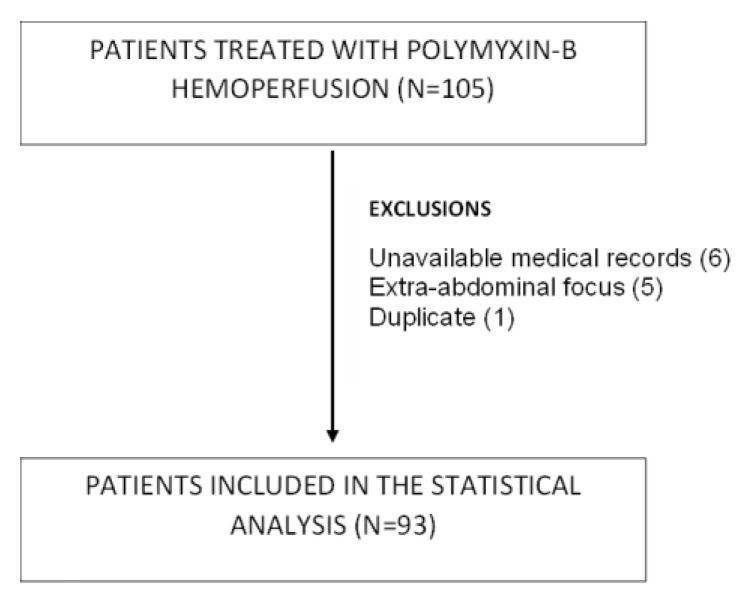
Patient selection flowchart.

**Figure 2 jpm-13-01023-f002:**
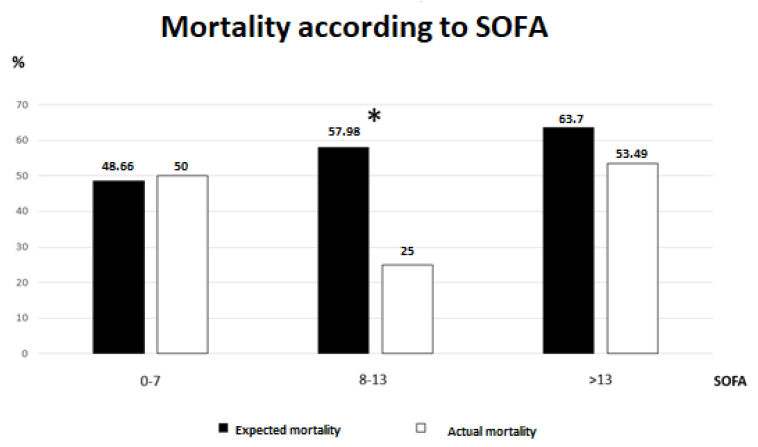
Mortality comparison in SOFA groups. The figure compares expected mortality according to APACHE II versus actual mortality, in the different SOFA subgroups. * *p* < 0.05.

**Table 1 jpm-13-01023-t001:** Patient characteristics.

VARIABLE	TOTAL	SURVIVORS	DECEASED	*p*
		FIRST MONTH	FIRST MONTH	
** *DEMOGRAPHICS* **				
**Sex (men)**	59 (63.4%)	38 (69.1%)	21 (55.3%)	
**Age (years)**	71.5	67.5 (59–76)	76.75 (64–82.1)	**0.014**
** *PROGNOSTIC SCORES* **				
**APACHE II**	28 (22.5–33.5)	28 (22–32)	31 (21.7–32.2)	0.431
**SOFA BEFORE PMX-HP**	13 (11–15)	13 (11–14)	14 (11–16)	0.079
**SOFA AFTER PMX-HP**	12 (9.5–14.5)	11 (7–13.2)	14 (11–16)	**0.001**
** *CORMOBIDITIES* **				
**Chronic kidney disease**	23 (24.7%)	9 (16.4%)	14 (36.8%)	**0.024**
**Chronic dialysis**	6 (6.5%)	1 (1.8%)	5 (13.2%)	**0.029**
**Renal transplant**	2 (2.2%)	1 (1.8%)	1 (2.6%)	0.79
**Hypertension**	51 (54.8%)	26 (47.3%)	25 (65.8%)	0.078
**Ischemic heart disease**	10 (7.5%)	6 (10.9%)	4 (10.5%)	0.953
**Peripheral vascular disease**	7 (7.5%)	3 (5.5%)	4 (10.5%)	0.362
** *SEPSIS TYPE* **				
**Localization**	UPPER 7 (7.5%)	UPPER 3 (5.5%)	UPPER 4 (10.5%)	0.233
	LOWER 61 (65.6%)	LOWER 35 (63.6%)	LOWER 26 (68.4%)	
	BILIAR 20 (21.5%)	BILIAR 15 (27.3%)	BILIAR 5 (13.2%)	
**Cause of abdominal sepsis**	Bowel Perforation	Bowel Perforation	Bowel Perforation	0.168
	22 (23.7%)	14 (25.5%)	8 (21.1%)	
	Dehiscence 33 (35.5%)	Dehiscence 24 (43.6%)	Dehiscence 9 (23.7%)	
	Intestinal ischemia 18 (19.4%)	Intestinal ischemia 7 (12.7%)	Intestinal ischemia 11 (28.9%)	
	Bowel obstruction 5 (5.4%)	Bowel obstruction 3 (5.5%)	Bowel obstruction 2 (5.3%)	
	Inflammation 11 (11.8%)	Inflammation 6 (10.9%)	Inflammation 5 (13.2%)	
	Others 4 (4.3%)	Others 1 (1.8%)	Others 3 (7.9%)	
**Source control**	Surgery 83 (89.2%)	Surgery 49 (89.1%)	Surgery 34 (89.5%)	0.96
	Drain 3 (3.2%)	Drain 2 (3.6%)	Drain 1 (2.6%)	
	None 7 (7.5%)	None 4 (7.3%)	None 3 (7.9%)	
**Time of evolution**	<24 H: 76 (81.7%)	<24 H: 44 (80%)	<24 H: 32 (84.2%)	0.606
	>24 H: 17 (18.3%)	>24 H: 11 (20%)	>24 H: 6 (15.8%)	
**Microbial cultures**	Yes 54 (58.1%)	Yes 34 (61.8%)	Yes 20 (52.6%)	0.377
**Type of pathogen**	Gram-positive coccus 2 (2.2%)	Gram-positive coccus 1 (1.8%)	Gram-positive coccus 1 (2.6%)	
	Gram-negative coccus 1 (1.8%)	Gram-negative coccus 1 (1.8%)	Gram-negative coccus (0%)	
	Gram-negative bacilli 27 (29%)	Gram-negative bacilli 16 (29.1%)	Gram-negative bacilli 11 (28.9%)	
	Fungus 1 (1.1%)	Fungus 1 (1.8%)	Fungus 0 (0%)	
	Polymicrobial flora (includes Gram-negative bacilli) 18 (19.4%)	Polymicrobial flora (includes Gram-negative bacilli) 12 (21.8%)	Polymicrobial flora (includes Gram-negative bacilli) 6 (15.8%)	
	Polymicrobial flora (without Gram-negative bacilli) 5 (5.4%)	Polymicrobial flora (without Gram-negative bacilli) 2 (3.6%)	Polymicrobial flora (without Gram-negative bacilli) 3 (7.9%)	
** *ANTIMICROBIALS* **				
**Antibiotics**	93 (100%)	SI 55 (100%)	SI 38 (100%)	
**Number of antibiotics**	One: 41 (44.1%)	One: 22 (40%)	One: 19 (50%)	0.613
	Two: 40 (43%)	Two: 25 (45.5%)	Two: 15 (39.5%)	
	Three: 12 (12.9%)	Three: 8 (14.5%)	Three: 4 (10.5%)	
**Type of antibiotics**	Cephalosporins 1 (1.1%)	Cephalosporins 1 (1.8%)	Cephalosporins 0 (0%)	
	Monobactams 1 (1.8%)	Monobactams 1 (1.8%)	Monobactams 0 (0%)	
	Carbapenems 82 (88.2%)	Carbapenems 47 (85.5%)	Carbapenems 35 (92.1%)	
	Aminoglycosides 4 (4.3%)	Aminoglycosides 3 (5.5%)	Aminoglycosides 1 (2.6%)	
	Quinolones 2 (2.2%)	Quinolones 1 (1.8%)	Quinolones 1 (2.6%)	
	Glycopeptides 29 (31.2%)	Glycopeptides 20 (42.5%)	Glycopeptides 11 (28.9%)	
	Metronidazole 10 (10.8%)	Metronidazole 3 (12.8%)	Metronidazole 2 (5.3%)	
	Linezolid 13 (14%)	Linezolid 7 (12.5%)	Linezolid 6 (15.8%)	
	Tigecycline 2 (3.6%)	Tigecycline 2 (3.6%)	Tigecycline 0 (0%)	
	Piperacillin-tazobactam 7 (7.5%)	Piperacillin-tazobactam 4 (7.3%)	Piperacillin-tazobactam 3 (7.9%)	
**Antifungal**	Yes 58 (62.4%)	Yes 32 (58.2%)	Yes 26 (68.4%)	0.37
**Type of antifungal**	Azoles 5 (5.4%)	Azoles 2 (3.6%)	Azoles 3 (7.9%)	0.475
	Echinocandins 53 (57%)	Echinocandins 30 (54.5%)	Echinocandins 23 (60.5%)	
** *VASOACTIVE AND INOTROPIC DRUGS* **				
**Noradrenaline**	93 (100%)	55 (100%)	38 (100%)	
**Noradrenaline dose**	High 66 (71%)	High 37 (67.3%)	High 29 (76.3%)	0.757
	Medium 19 (20.4%)	Medium 11 (20%)	Medium 8 (21.1%)	
	Low 4 (4.3%)	Low 3 (5.5%)	Low 1 (2.6%)	
**Adrenaline**	23 (24.7%)	12 (21.8%)	11 (28,9%)	0.433
**Other drugs**	19 (20.4%)	10 (18.2%)	9 (23.7%)	0.518
** *RENAL FAILURE* **				
**Oliguria**	67 (72%)	34 (61.8%)	33 (86.8%)	**0.008**
**CVVH**	58 (62.4%)	28 (50.9%)	30 (78.9%)	**0.006**
**Renal failure (before hemoperfusion)**	66 (71%)	37 (67.3%)	29 (76.3%)	0.413
**Renal failure (after hemoperfusion)**	60 (64.5%)	32 (58.2%)	28 (73.7%)	0.125

**Table 2 jpm-13-01023-t002:** Variation in analytical results and SOFA, before and after hemoperfusion, for surviving and deceased patients.

SURVIVORS AFTER ONE MONTH				
	BEFORE HEMOPERFUSION	AFTER HEMOPERFUSION	% INCREASE	*p*
Leukocytes (×103/mL)	13,250	16,000	17.18	0.595
Platelets (×103/mL)	155,000	79,000	−96.20	<0.001
Haemoglobin (gr/dL)	10	9	−11.11	<0.001
INR	1.28	1.08	−18.52	<0.001
Fibrinogen (mg/dL)	667	710	6.056	0.609
Creatinine (mg/dL)	2.02	1.43	−41.26	0.001
pH	7.3	7.37 *	0.95	<0.001
Bicarbonate (mmol/L)	21	24 *	12,5	<0.001
GOT (UI/L)	52.5	50.5	−3.96	0.207
GPT (UI/L)	39	47.5	17.89	0.855
Arterial lactate(mmol/L)	3.5	1.6 *	−118.75	<0.001
C reactive protein (mg/ dL)	21.35	10.85	−96.77	<0.001
Procalcitonin (ng/dL)	6.75	2.57	−162.65	<0.001
**DECEASED AFTER ONE MONTH**				
	**BEFORE HEMOPERFUSION**	**AFTER HEMOPERFUSION**	**% INCREASE**	** *p* **
Leukocytes (×10^3^/mL)	12,450	17,500	28.857	0.083
Platelets (×10^3^/mL)	150,000	69,000	−117.39	<0.001
Haemoglobin (gr/dL)	10.1	8.7	−16.09	<0.001
INR	1.41	1.12	−25.89	0.136
Fibrinogen (mg/dL)	648	588	−10.20	0.168
Creatinine (mg/dL)	2.07	1.71	−21.05	0.086
pH	7.3	7.34 *	0.54	0.066
Bicarbonate (mmol/L)	19	22 *	13.64	0.025
GOT (UI/L)	58	92 *	36.96	0.174
GPT (UI/L)	37	70	47.14	0.084
Arterial lactate(mmol/L)	4.5	2.85 *	−57.9	0.077
C reactive protein (mg/ dL)	25	12.9	−93.8	0.022
Procalcitonin (ng/dL)	13.3	11.43	−16.36	0.167
SOFA	14	14	0.00	0.556

**Table 3 jpm-13-01023-t003:** Multivariant analysis of mortality risk factor son the first 30 days. Logistic regression results.

VARIABLE	ODDS RATIO	IC 95%	
Variation on lactate levels	1.22	1.006–1.48	0.043
Chronic Kidney Disease	3.646	1.02–9	0.045
Discontinuation of vasoactive drugs	0.24	0.09–0.63	0.004

## Data Availability

No publicly available dataset was generated during the study period.

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
