# Peer review of "Mortality Risk Prediction in Abdominal Septic Shock Treated with Polymyxin-B Hemoperfusion: A Retrospective Cohort Study"

_jpm, 2023, doi:10.3390/jpm13071023_

Round 1

Reviewer 1 Report

General comments

The MS No.: [JPM] Manuscript ID_ jpm-2437719: Mortality risk prediction in abdominal septic shock treated with polymyxin-B hemoperfusion: A retrospective cohort study. by authors: Sergio Garcia-Ramos, Estrela Caamano, Patrocinio Rodriguez, Pilar Benito, Alberto Calvo, Silvia Ramos, Mercedes Power, Ignacio Garutti and Patricia Piñeiro, represent a retrospective, observational, single-centre cohort study of 93 patients suffering from abdominal septic shock treated with polymyxin-B hemoperfusion (PMX-HP). Authors assessed the data before and after PMX-HP and concluded that in in patients with abdominal septic shock, the addition of PMX-HP to the standard therapy resulted in lower mortality than expected.

The title of MS is clear and adequate, but I suggest a small correction (see text).

Abstract is well written, clear and and self-explanatory for the readers.

The introduction section is well written and explains the basic concepts of sepsis, PMX and aims of the study.

Materials and methods: Classic retrospective, observational, cohort study that includes patients who were treated with PMX-HP. Statistical analysis is appropriate, but It would be very good to do a ROC analysis to see the effects of the therapy based on the value of the AUC.

Results: Table 1 is not well organized in one of its parts. The laboratory results before hemoperfusion section should be right next to the laboratory results after hemoperfusion section, and each parameter before and after treatment should be statistically compared to see the effects of therapy, for example:

The laboratory results before hemoperfusion

The laboratory results after hemoperfusion

P value

Leukocytes (x103/mL)

Platelets (x103/mL)

Haemoglobin

INR

In summary, the manuscript is based on the fact that some meta-analyses have suggested that PMX-HP could be a useful treatment for patients with sepsis. Obtained results demonstrates that PMX shows good results in the treatment of abdominal septic shock and reduces patient mortality.

Authors also stated that their study had several limitations, for example the absence of a control group as a result of the use of PMX-HP as standard care in this context since 2015.

Considering that sepsis is a major medical problem that is difficult to diagnose and treat and has an unpredictable prognosis, I think that any attempt to solve this problem is valuable. Finally, the article represents a contribution to the overall scientific knowledge in this area and provides a solid basis for further analyses.

Specific comments are given in the text.

Having all the above in mind, I suggest to the editor to accept this review manuscript for publication after minor revisions.

My final opinion: acceptable for publication after minor revisions.

Author Response

We changed the contents of table 2, comparing surviving and deceased patients, as reviewer 2 also suggested. 

We have added a paragraph explaining the table contents below it. 

Reviewer 2 Report

 This paper is interesting, well written, however in the Results the presentation is difficult to read.

1) To clarify and highlight the results, I suggest changing the Table 1, part "Laboratory results before / after hemoperfusion", in a unified bar graphs, showing the significant statistical values  

2) In Table 2 " Variation in analytical results and SOFA, before and after hemoperfusion", for me some results are not clear. To give you an example, for the factor "platelets" in Decease group a variation of 117.39%, what does it mean? Please explain these data with an appropriate legend

the english language is acceptable

Author Response

We have changed the contents of table 2, comparing surviving and deceased patients, following both of the reviewers suggestion. 

Round 2

Reviewer 2 Report

I have no further comments

good quality